# Genetic diversity of SARS-CoV-2 infections in Ghana from 2020-2021

Collins M. Morang'a [1,2], Joyce M. Ngoi[1,2], Jones Gyamfi[3], Dominic S. Y. Amuzu[1,2], Benjamin D. Nuertey [4], Philip M. Soglo[1,2], Vincent Appiah[1,2], Ivy A. Asante[5], Paul Owusu-Oduro[6], Samuel Armoo[7], Dennis Adu-Gyasi [8], Nicholas Amoako[1,8], Joseph Oliver-Commey[9], Michael Owusu [10], Augustina Sylverken[10], Edward D. Fenteng[11], Violette V. M'cormack[1,2], Frederick Tei-Maya [1,2], Evelyn B. Quansah[1,2], Reuben Ayivor-Djanie [3], Enock K. Amoako[1,2], Isaac T. Ogbe[1,2], Bright K. Yemi[1,2], Israel Osei-Wusu [1,2], Deborah N. A. Mettle[1,2], Samirah Saiid[1,2], Kesego Tapela[1,2], Francis Dzabeng[1,2], Vanessa Magnussen[1,2,5], Jerry Quaye [1,2], Precious C. Opurum[1,2], Rosina A. Carr [3], Patrick T. Ababio[11], Abdul-Karim Abass[12], Samuel K. Akoriyea[13], Emmanuella Amoako[14], Frederick Kumi-Ansah [14], Oliver D. Boakye[15], Dam K. Mibut[4], Theophilus Odoom[15], Lawrence Ofori-Boadu[16], Emmanuel Allegye-Cudjoe[17], Sylvester Dassah[18], Victor Asoala[18], Kwaku P. Asante[8], Richard O. Phillips[10], Mike Y. Osei-Atweneboana[7], John O. Gyapong[3], Patrick Kuma-Aboagye[19], William K. Ampofo[5], Kwabena O. Duedu [3], Nicaise T. Ndam[20], Yaw Bediako[1,2,21], Peter K. Quashie [1,2✉], Lucas N. Amenga-Etego [1,2✉] & Gordon A. Awandare [1,2✉]

The COVID-19 pandemic is one of the fastest evolving pandemics in recent history. As such, the SARS-CoV-2 viral evolution needs to be continuously tracked. This study sequenced 1123 SARS-CoV-2 genomes from patient isolates (121 from arriving travellers and 1002 from communities) to track the molecular evolution and spatio-temporal dynamics of the SARS-CoV-2 variants in Ghana. The data show that initial local transmission was dominated by B.1.1 lineage, but the second wave was overwhelmingly driven by the Alpha variant. Subsequently, an unheralded variant under monitoring, B.1.1.318, dominated transmission from April to June 2021 before being displaced by Delta variants, which were introduced into community transmission in May 2021. Mutational analysis indicated that variants that took hold in Ghana harboured transmission enhancing and immune escape spike substitutions. The observed rapid viral evolution demonstrates the potential for emergence of novel variants with greater mutational fitness as observed in other parts of the world.

A full list of author affiliations appears at the end of the paper.

A year after the World Health Organization (WHO) declared the coronavirus disease 2019 (COVID-19) caused by severe acute respiratory syndrome coronavirus 2 (SARS-CoV-2) a pandemic, over 190 million confirmed cases and 4 million deaths have been reported worldwide[1]. As of 27th September 2021, Ghana's cumulative COVID-19 cases stood at 127,482 with an active case count of 3088 and 1156 deaths[2]. Through the COVAX initiative, 1.23 million people in Ghana have received vaccine doses, with 376,000 fully vaccinated with the AstraZeneca vaccine[3,4] Compliance to prescribed preventive measures have been low to average within the communities, and there is evidence of persistent albeit mostly asymptomatic infections, especially in Accra and other major cities in Ghana[5].

COVID-19 control measures in Ghana have evolved with the global pandemic. Ghana's international airports and all land borders were closed to international travel on 22nd March 2020, followed by a partial lockdown of two major cities from March 30th to 22nd April 2020. This was coupled with enhanced testing and contact tracing to track community spread. The airport was reopened to international travel on 1st September 2020, with twofold containment measures; (1) travellers must show proof of a negative COVID-19 test (taken at most 72 h before arrival) and (2) travellers must be negative for the SARS-CoV-2 antigen test upon arrival at the Kotoka International Airport (KIA)[2]. The guidelines for travellers who test positive at the airport have changed over time. Initially, travellers who tested positive upon arrival had to undergo mandatory isolation for at least 14 days (at travellers' cost) and were only allowed to go after a negative PCR/antigen test. The guidelines were later relaxed to allow self-isolation, but mandatory isolation has been reinstated due to poor compliance[2]. Currently, a positive test leads to a minimum of 3 days in isolation. After 3 days, travellers who test negative by RT-PCR after 3 days are allowed to leave isolation. Since January 2021, all positive samples from travellers have been made available for genomic sequencing.

Like other RNA viruses, most mutations in the SARS-CoV-2 genome arise during viral replication, and the resulting mutant viruses are then subjected to selective pressures within the host and/or during inter-person transmission. Whole-genome sequencing (WGS) is critical in tracking viral genomic changes and may help to understand phenotypic changes. According to the WHO, several variants of interest (VOIs) have been shown to harbour amino acid changes associated with enhanced community transmission or multiple COVID-19 cases/clusters in numerous countries[6,7]. Other VOIs have proven to be variants of concern (VOCs) due to their increased transmissibility, virulence, and disease severity, or decreased susceptibility to public health measures, available diagnostics, vaccines, and therapeutics[7]. These may be demonstrated by increased receptor binding, reduced virus neutralisation by antibodies generated against previous infection or vaccination, loss or reduced diagnostic detection, or increased replication[7]. The SARS-CoV-2 VOCs and VOIs listed by WHO include; Alpha (B.1.1.7), Beta (B.1.351), Gamma (P.1), Delta (B.1.617.2), Epsilon (B.1.427/B.1.429), Zeta (P.2), Eta (B.1.525), Theta (P.3), Iota (B.1.526) and Kappa (B.1.617.1)[7]. These variants were first documented by the UK, Brazil, India, USA, and Philippines[6,7]. Only Beta and Eta were first reported in South Africa and Nigeria, respectively[6,8].

Having established a local capacity for sequencing and analysing SARS-CoV-2 genomes in Ghana[9], molecular surveillance has continued using samples provided by the COVID-19 testing laboratories across the country. In addition, some samples from international travellers who tested positive on arrival at the airport were also analysed to track the introduction of new variants into the country. Thus, this report provides a comprehensive analysis of the genetic diversity of SARS-CoV-2 viruses that caused infections in the communities in Ghana from March 2020 to September 2021.

## Results

### Demographic and clinical characteristics of study participants.
A total of 2213 samples were availed for WGS, 1987 samples were processed for WGS, and 1573 samples made it to lineage assignment (Supplementary Table 1). From the 1573 samples, 31% (496/1573) had frameshift mutations on several viral proteins such as the ORF7a/b, ORF8, ORF6, ORF1a/b, E, S, N, and ORF3a. Sixty-nine percent (1077/1573) of these sequences have been submitted to the GISAID and European Nucleotide Archive (ENA) (Supplementary Data 1), and are analysed herein alongside 46 genomes from Ngoi et al.; the rest (496/1573) are under investigation before submission to GenBank/GISAID. Thus, a total of 1123 COVID-19 PCR positive samples from ten regions in Ghana between March 2020 and September 2021 were successfully sequenced. Of these, 45% (500/1123) were near full-length genomes (missingness ($N$) < 500), 48% (539/1123) had 500–4000 unresolved nucleotide assignments ($N$), and only 7% (84/1123) had 4000–8913 missingness ($N$). Only 19% (215/1123) of all the samples were collected in 2020, while the remaining were collected in 2021 ($n = 908/1123$). Of the 1123 samples, 89% ($n = 1002$) were community samples while 11% ($n = 121$) were from travellers arriving in the country through the KIA. Sampling contribution from the different regions in Ghana, was as follows: Ashanti (4%, $n = 47$), Bono East (1%, $n = 13$), Central (12%, $n = 133$), Eastern (5%, $n = 52$), Greater Accra (44%, $n = 490$), Northern (5%, $n = 55$), Upper East (<1%, $n = 4$), Upper West (1%, $n = 12$), Volta (9%, $n = 103$) and Western (8%, $n = 93$) (Fig. 1a). With the highest number of active COVID-19 cases (GHS, 2021), the Greater Accra region had the highest number of sequenced samples (Supplementary Table 2).

There were more males (56%, 561/1002) compared to females (42%, 418/1002), with only a few missing entries (2%, 23/1002). Most of the study participants were between 21–40 years (54%, $n = 507$) as compared to the other categories; ≤20 (11%, $n = 104$), 41–60 (24%, $n = 230$), and 61+ (11%, $n = 106$) years (Supplementary Table 3). Although most of the samples were from asymptomatic individuals who reported for COVID-19 testing, a good number of genomes (121/1002) were from individuals who presented at treatment centres with mild/moderate or severe/critical symptoms. Majority of the individuals classified as mild/moderate (69.4%, 84/121) presented with fever/chills, cough, pains, sore throat, diarrhoea, runny nose, nausea/vomiting, loss of smell, loss of taste or headache. Our data show that less than a third of individuals presenting at the hospital were classified as severe/critical (30.5%, 37/121) with difficulty breathing (shortness of breath), hypoxia or multiorgan system dysfunction.

### Genomic epidemiology of variants circulating in Ghana.
Overall, the Delta lineages (32%, 360/1123), Alpha (20%, 224/1123), B.1.1.318 (16%, 180/1123), B.1.1 (9%, 106/1123), and Eta (4%, 47/1123) made up the top viral lineages within the sequenced SARS-CoV-2 genomes in Ghana over the period (Fig. 1b). Since the B.1.1.318 had the third-highest frequency, it is considered a variant under monitoring in Ghana. In 2020, the Ashanti, Central, Eastern, Greater Accra and Western regions had the highest numbers of confirmed cases of COVID-19 in Ghana. Different regions showed variations in relative frequencies of the variants. In 2020, genomes which clustered within the B.1.1 lineage dominated samples from Greater Accra (37%, 15/41), Volta region (56%, 10/18), Western region (51%, 23/45) and Central region (45%, 22/49). B.1.623 (28%, 5/18) co-dominated with B.1.1 (28%, 5/18) in the

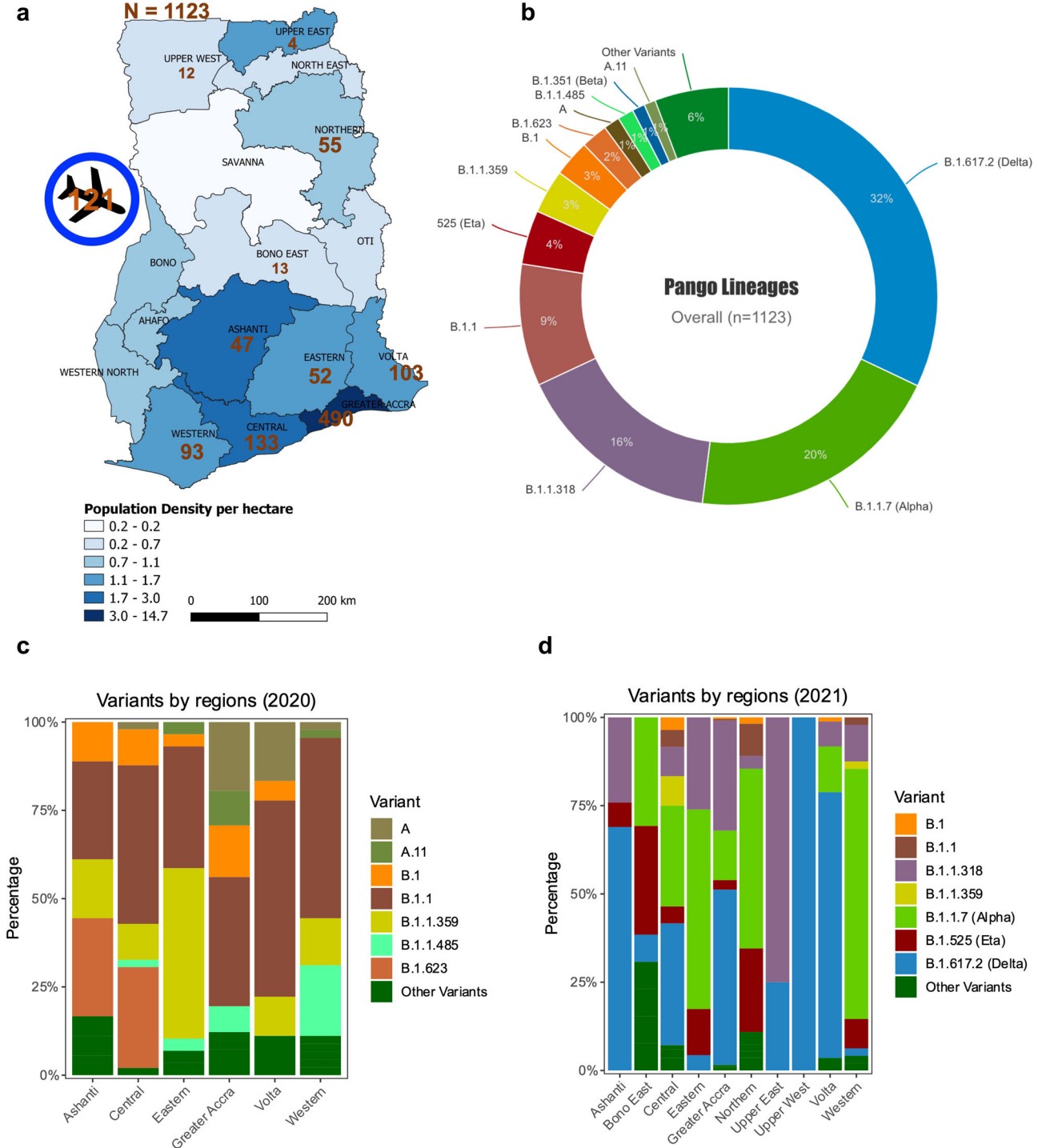

**Fig. 1 Overall genomic epidemiology of the circulating variants across various regions in Ghana (N = 1123). a** Map showing the ten sampled regions (n = 1002) and travellers arriving in the country (121) at the Kotoka International Airport in the Greater Accra Region. The colour scheme shows the population density per hectare in each region, while the numbers are samples sequenced per region. **b** Overall SARS-CoV-2 lineages in Ghana for both local participants and travellers (n = 1123); the lineages are represented by different colours, while the percentages represent the proportion of cases with that particular lineage out of 1123 participants. Lineages with <6 participants are represented as "Other Variants". **c** Out of the 1002 samples sequenced from community samples, 200 samples were sequenced in 2020 and 802 samples sequenced in 2021. The stacked plot shows the percentage distribution of lineages in six regions in 2020. Lineages are represented in unique colours as indicated on the plots. **d** The stacked plot shows the percentage distribution of lineages in 2021 from ten regions across Ghana (n = 802).

Ashanti region whereas B.1.1.359 (48%, 14/29) dominated alongside B.1.1 (35%, 10/29) in the Eastern region (Fig. 1c).

In 2021, there was a marked shift in the circulating variants and occurrence of regional specific outbreaks, with Eta dominating in Northern and middle belt regions, while B.1.1.318 dominated the major cities. The highest frequencies of Eta variants were observed in the Northern (24%, 13/55), Bono East (31%, 4/13) and Eastern (13%, 3/23) regions (Fig. 1d). The

city of Tamale in the Northern region is the gateway, and central trading hub with Ghana's northern neighbours, whilst the Bono East region harbours major interaction routes with Ivory Coast in the western corridor of Ghana. Meanwhile, nearly a third of all the variants detected in 2021 were B.1.1.318 (22%, 176/802), and Greater Accra, where the capital city and the major international airport are located, had 80% (140/176) of all the B.1.1.318 genomes (Fig. 1d). These data suggest that Eta and B.1.1.318 variants, which dominated transmission in these areas in April–May 2021, could have been introduced through these major land borders. The B.1, B.1.1.359, B.1.1, and B.1.623 that dominated Ghana in 2020 became supplanted by Alpha and Delta VOCs in most of the regions. It is worth noting that in regions where more than 50 samples were sequenced in 2021, there was penetration or transmission of various VOCs, including the Central region, Bono East, Greater Accra, and Volta Region (Fig. 1d).

**Importation of SARS-CoV-2 variants into Ghana by travellers**. One hundred and twenty-one of the sequenced samples (11%, 121/1123) were obtained from travellers identified as COVID-19 positive at the KIA. Of this number, Alpha accounted for 39% ($n = 47$) of the genomes while the other VOCs accounted for lower proportions; Beta (6%, $n = 7$), and Delta lineages (7%, $n = 8$) (Supplementary Table 4). The VOIs such as Eta (4%, $n = 5$), and the local variant under monitoring, B.1.1.318, 3% ($n = 4$) were detected at low proportions (Supplementary Table 4). Importantly, the VOC Alpha was identified in travellers entering Ghana from all over the World, including other African countries, in January and March 2021 (Table 1). Furthermore, VOCs were detected in travellers from several of Ghana's neighbouring countries including, Nigeria, Ivory Coast, and Burkina Faso, demonstrating that these variants were already in those countries even though not reported or detected (Table 1). In most cases, VOCs and VOIs were identified amongst quarantined travellers before their detection within local samples. Travellers from Nigeria, Dubai, and the UK accounted for most detections of Alpha, Beta, Eta, and Delta variants (Table 1). Interestingly, the Beta and Kappa variants did not become dominant in Ghana; instead, B.1.1.318, which was detected in travellers from Nigeria, Gabon, and Dubai, became dominant in Ghana between April and June 2021.

**Temporal trends of SARS-CoV-2 variant detection and frequency**. Ghana was one of the last African countries to detect COVID-19 cases in March 2020, and the waves of COVID-19 in Ghana have lagged slightly behind other African countries and significantly behind the rest of the World (Fig. 2a). Previous work from our group described the viral genome dynamics between March and May 2020, when Ghana was largely closed to international travel (Ngoi et al.[9]). Different variants rose to dominance at different times and during different infection waves across the country (Fig. 2a, b). Variants that cluster closely to B.1.1 were first detected in March 2020 (50%, 2/4) and peaked in July 2020 (59%, 27/46). The B.1.1 lineage remained dominant until November 2020 (55%, $n = 16/29$). In September 2020, the B.1.1.359 (56%, $n = 10/18$) became dominant while in December 2020, the B.1.623 (33%, $n = 9/27$) was dominant alongside B.1.1 (30%, $n = 8/27$) (Fig. 2b). The Alpha VOC, first detected in local samples from 1st January 2021, quickly supplanted the B.1.1 lineage as the most dominant circulating lineage (72%, $n = 102/141$) in January 2021. Alpha maintained high frequency (60%, $n = 42/70$) in February 2021, followed by the VOI, Eta (20%, $n = 14/70$) (Fig. 2b, c). Alpha was still dominant in March 2021 (71%, 22/31) but was almost completely supplanted in April 2021, as B.1.1.318 rose to prominence. Sequences clustering to B.1.1.318

had been detected as early as February 2021 (4%, $n = 3/70$) but became dominant in April 2021 (63%, $n = 38/60$) alongside Eta (23%, $n = 14/60$). May 2021 represented the peak of B.1.1.318 (82%, $n = 79/96$) and the emergence of the Delta lineages (5%, $n = 5/96$) in local circulation. Delta lineages increased in proportion over time, overtaking the B.1.1.318 in June 2021. Delta lineages surged in July and dominated with a significant presence in Ghana as at September 2021 (Fig. 2c).

**Genetic diversity and evolutionary relationships of the SARS-CoV-2 variants**. Amongst the many individual lineages represented in the data presented here, Delta lineages, Alpha, B.1.1.318, B.1.1.359, B.1.1, and Eta were the most evolved, with the highest genetic diversity (Fig. 3a). These variants exhibited a variation in the number of mutations from sample to sample, with Delta, Alpha and B.1.1.318 presenting a mean ~30 (spread/ range of 20–45) mutations in the majority of the genomes (Fig. 3a). The Delta VOCs had the highest mean (~35) and presented an interquartile range of mutations from 25 to 45 mutations across all the samples (Fig. 3a). It is worth noting that this level of genetic diversity in Delta lineages was mainly attributed to the sublineages ($n = 200/360$); AY.39 (174/200) and AY.37 (15/200) lineages (Supplementary Table 5). Most of the other lineages with a small range of mutations were reported in 2020 and occurred spontaneously in very few samples hence the relatively low genetic diversity (Fig. 3a). The high level of genetic diversity in most VOCs, including the B.1.1318, is probably indicative of Ghana's local evolution and consequential adaptation compared to the other variants that did not gain prominence in the Ghanaian population (Fig. 3a).

A snapshot of the evolutionary relationship of these VOCs in Ghana shows a relationship of variants through space and time throughout the epidemic (Fig. 3b). Using a phylogenetic tree, we outline the phylogenetic relationships of VOCs and how they gained prominence coinciding with the COVID-19 waves in Ghana. The outbreak of the COVID-19 pandemic started in mid-November 2019, but then the tree shows that the earliest lineages in Ghana are dated March 2020, although most VOCs were introduced in 2021 (Fig. 3b). The phylogenetic analysis of the genomes from Ghana shows similarities to VOCs around the World, with all the VOCs having the same common ancestor (Wuhan). Still, as they diverge, they share uniquely more recent ancestors; for example, we show that the B.1.1.318 and Alpha variants share several recent ancestors (Fig. 3b). The root-to-tip divergence of the VOCs as a function of sampling time show a molecular clock of the various VOCs, and with strong evidence, the variants are evolving in a clocklike manner ($R^2 = 0.71$) (Fig. 3c). The variants in Ghana are gaining ~26 mutations per year, and of particular interest is the B.1.1.318 that did not gain prominence worldwide, but its molecular clock is similar to most of the VOCs in Ghana (Fig. 3c). Mutational fitness of the B.1.1.318 lineage showed that ten samples had spike mutations that were likely to confer viral fitness (mutational fitness > 1) (Fig. 3d).

**Mutational analysis of the amino acid substitutions**. The most abundant substitution in all the samples was the spike D614G (97%, 972/1002), followed by ORF1b: P314L (91%, 915/1002) (Fig. 4a). For most of the genes, one or more amino acid substitutions occurred in more than 100 samples, although spike protein dominated the profile (Fig. 4a). Interestingly, some variants with different evolutionary lineages had similar amino acid substitutions, mainly spike glycoprotein. The Eta variant had the highest (three) individual amino acid substitutions (Q52R, Q677H and F888L) in the spike protein compared to other VOCs,

**Table 1 Transmission of SARS-CoV-2 variants into Ghana from other countries (n = 121).**

| Travel history | Mar 2020 variant (n) | n | Jan 2021 variant (n) | n | Mar 2021 variant (n) | n | Jun 2021 variant (n) | n | Sub total |
|---|---|---|---|---|---|---|---|---|---|
| Burkina Faso | | 0 | Alpha(1) | 1 | Beta(1) | 1 | | 0 | 2 |
| Cote d'Ivoire | | 0 | | 0 | Alpha(3) | 3 | | 0 | 3 |
| Dubai | | 0 | A.23.1 (1), B.1.1 (1), Alpha(4) | 6 | B.1.1.318 (1), Alpha(1), Beta(1) | 3 | | 0 | 9 |
| Egypt | | 0 | | 0 | C.36.3 (1) | 1 | | 0 | 1 |
| Emirates | | 0 | | 0 | Alpha(1), Kappa(1) | 2 | | 0 | 2 |
| Gabon | | 0 | | 0 | B.1.1.318 (1) | 1 | | 0 | 1 |
| The Gambia | | 0 | | 0 | Alpha(1) | 1 | | 0 | 1 |
| Guinea | | 0 | | 0 | Alpha(1) | 1 | | 0 | 1 |
| Hungary | A.11 (1) | 1 | | 0 | | 0 | | 0 | 1 |
| India | B.1 (1) | 1 | B.1.36.8 (1) | 1 | | 0 | | 0 | 2 |
| Ivory Coast | | 0 | B.1 (2), Alpha(1) | 3 | | 0 | | 0 | 3 |
| Kenya | | 0 | | 0 | A.23.1 (1), Alpha(2) | 3 | | 0 | 3 |
| Lebanon | | 0 | Alpha(1) | 1 | Alpha(4) | 4 | | 0 | 5 |
| Liberia | | 0 | B.1.1 (1), R.1 (3) | 4 | | 0 | | 0 | 4 |
| Mali | | 0 | A.21 (1) | 1 | | 0 | | 0 | 1 |
| Namibia | | 0 | | 0 | Beta(1) | 1 | | 0 | 1 |
| Netherlands | | 0 | B.1.177.81 (1) | 1 | | 0 | | 0 | 1 |
| Nigeria | | 0 | B.1.1 (1), B.1.1.10 (1), Alpha(7), B.1.177.86 (1), Eta(4) | 14 | Beta(1), B.1.1.318 (1) | 2 | | 0 | 16 |
| Norway | B.1.1 (1) | 1 | | 0 | | 0 | | 0 | 1 |
| Senegal | | 0 | B.1.1.420 (1) | 1 | | 0 | | 0 | 1 |
| Sierra Leone | | 0 | B.1.1 (1) | 1 | | 0 | | 0 | 1 |
| South Korea | | 0 | | 0 | Alpha(1) | 1 | | 0 | 1 |
| Tanzania | | 0 | Alpha(2), Beta(1) | 3 | | 0 | | 0 | 3 |
| Turkey | | 0 | | 0 | Alpha(2) | 2 | | 0 | 2 |
| UK | B (1), B.1 (1) | 2 | Alpha(2), B.1.177.7 (1), Eta(1) | 4 | | 0 | | 0 | 6 |
| UK, USA, Dubai | B.1 (1) | 1 | | 0 | | 0 | | 0 | 1 |
| Ukraine | | 0 | | 0 | Alpha(1) | 1 | | 0 | 1 |
| USA | | 0 | B.1.243 (1) | 1 | Alpha(2), B.1.526 (Iota) (2) | 4 | | 0 | 5 |
| Not reported | A (1), A.11 (3), B.1 (2), B.1.1 (1), B.1.220 (1), B.40 (1) | 9 | B.1 (3), B.1.1 (3), B.1.1.409 (1), Alpha(10), B.1.177 (2), B.1.177.86 (1), Beta(2), L.3 (2) | 24 | | 0 | B.1.1.318 (1), B.1.617.2 (Delta) (8) | 9 | 42 |
| Total | | 15 | | 66 | | 31 | | 9 | 121 |

probably contributing to its adaptability in Africa. Compared to other VOCs, the substitutions unique to the Alpha variant were S13I, R567K, A570D, and T716I. The only substitution unique to the B.1.1.318 on the spike protein was the D1127G compared to other VOCs and VOIs (Fig. 4b). Within these samples, the Delta lineages shared 14 substitutions (T19R, G142D, R158G, A222V, L452R, T478K, E484Q, D614G, S680F, P681R, D950N, K1191N, G1219V and C1253F) (Fig. 4b). The unique substitutions were fewer than shared amino acid substitutions among lineages (Fig. 4b), thus explaining the increased abundance of some substitutions among the VOCs. Those with the highest frequency in the spike protein among Delta lineages, B.1, B.1.1, B.1.1.318, Alpha, Beta and Eta were fitness substitutions D614G and P681R/H (Fig. 4b). Alpha and B.1.1.318 had the P681H substitution while P681R was present in Delta lineages. Immune escape substitution E484K was present in B.1.1.318, Beta, and Eta, while Delta variants frequently presented with E484Q substitution.

## Discussion

Having established local capacity to generate high-quality genome sequences and comprehensively analyse them in-house, we conducted genomic surveillance in Ghana from March 2020 to September 2021 and performed an in-depth analysis on the resulting 1123 SARS-CoV-2 sequences obtained. This study represents the most extensive genomic analysis of the SARS-CoV-2 viruses

driving the COVID-19 pandemic in Ghana. Trends in SARS-CoV-2 infections in Ghana have followed a similar pattern as the rest of Africa and globally, although Ghana has constantly lagged behind the rest of Africa and the World in the COVID-19 disease during the second and third waves. The emergence of new variants such as Alpha and Delta have been responsible for the second and third COVID-19 disease waves in Ghana, as has been reported in other countries in Africa and globally[1]. The Ghana Health Service monitoring data indicate that the Greater Accra Region was and still is the epicentre of COVID-19 infections[2]. Other regions with major urban cities, including Ashanti, Western and Central, have had high infection levels and similar circulating variants as Greater Accra, likely due to a high volume of intercity travel across these regions. Regions like Northern and Upper East, further from Accra, tended to have different variants during the second wave. In the third wave, these regions still lag behind the rest of the country and do not seem to be undergoing a third wave yet[2]. These regions experience much lower international travellers from global COVID-19 hotspots than Greater Accra and Ashanti. Furthermore, they have a more sparse population with less congested cities than Greater Accra and Ashanti regions. Studies have shown that higher population density increases contact rates necessary for SARS-CoV-2 disease transmission[10].

Before the airport reopened to international travellers in September 2020, the B.1.1 variant was dominant in Ghana and remained the most dominant circulating lineage throughout

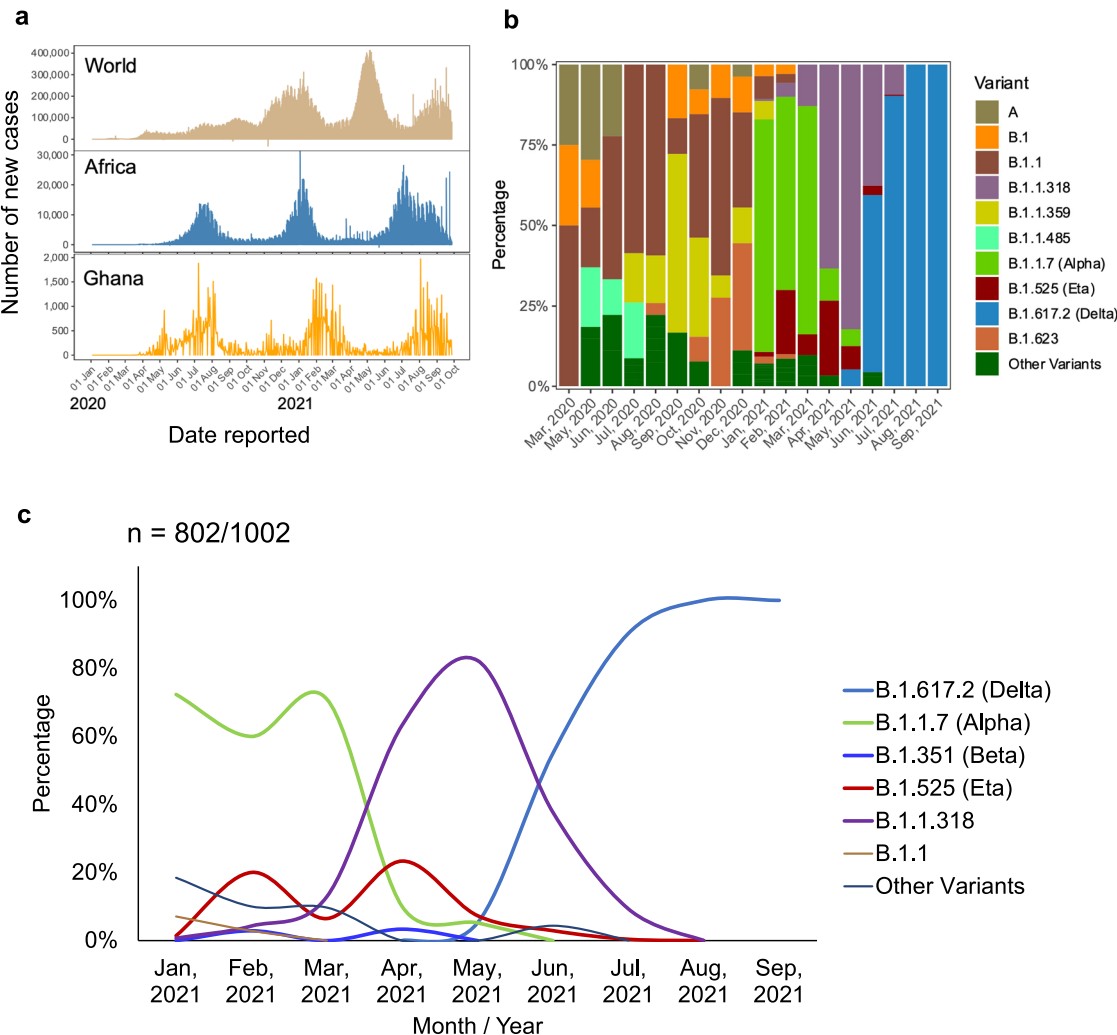

**Fig. 2 Trends in the variant epidemiology in Ghana. a** Trends of COVD-19 in the World, Africa and Ghana, which indicates the date reported against the number of new cases. This shows the number of confirmed COVID-19 cases recorded daily in the World, Africa, and Ghana from January 2020 to October 2021. The data were obtained from World Health Organisation (https://covid19.who.int/WHO-COVID-19-global-data.csv). **b** Trends in prevalence of major variants circulating in Ghana from March 2020 to September 2021. The Y-axis shows the percentage distribution (n = 1002/1123) of various variants including VOCs across the various months (X-axis) while the lineages are represented by different colours. **c** Impact of emergence of variants of concern (VOCs) on dynamics of viral lineages dominating community transmission in Ghana in 2021 (802/1002).

2020. In January 2021, the Alpha variant supplanted B.1.1 and became the leading cause of all reported cases nationally. Our analysis showed that major variants such as Alpha, Beta, Delta, Eta, and Kappa were detected in samples from arriving travellers before being observed in community cases. This was indicative that VOCs and VOIs were likely imported into Ghana through travellers from other countries, including African countries which had not reported such variants. These results also suggest that the delayed entry of VOCs into Ghana was at least partly due to the mandatory antigen testing on arrival at KIA (introduced when the borders were reopened in September 2020) and subsequent imposition of a policy for compulsory isolation of all antigen-positive passengers (in January 2021). The results also suggest that some individuals escaped detection and seeded local outbreaks. Alpha was the most frequent variant among travellers, followed by the Delta, Beta, Eta, and B.1.1.318, with other variants accounting for the rest. Due to the increased transmissibility of Alpha[11], it was not surprising that it became the most dominant variant in local transmission in major cities. Eta was primarily responsible for COVID-19 cases that we analysed from the Northern Region. Eta was first identified in the UK, and it

contains immune escape substitutions (del69-70 and E484K) and increased fitness substitutions (P681R)[12].

Though largely unheralded, B.1.1.318 had been responsible for the second wave in Mauritius[13]. Given the spread of B.1.1.318 in Ghana, Mauritius and other countries, it looked poised to become a dominant SARS-CoV-2 variant. From our dataset, B.1.1.318 appeared to replace Alpha as the predominant lineage in community infections in March 2021 and remained dominant until June 2021, when it was replaced by Delta lineages. The combination of spike substitutions T95I, E484K, D614G, P681H, and D796H in B.1.1.318 may have allowed it to escape Alpha-induced immunity in the Ghanaian population and replace the Alpha variant. This analysis further solidifies the evidence for E484K, L5F, A27S and D614G as spike substitutions that increase infectiousness with minimal impacts on severity[14].

Our dataset showed that Delta lineages dominated in Ghana from June 2021, and remained dominant as at September 2021. The worldwide dominance of Delta variants[15,16] has been linked to the P681H/R substitutions[3] as well as additional mutations in the viral RNA-dependent RNA-polymerase coding sequence. It is opined that these mutations enhance replication speed and

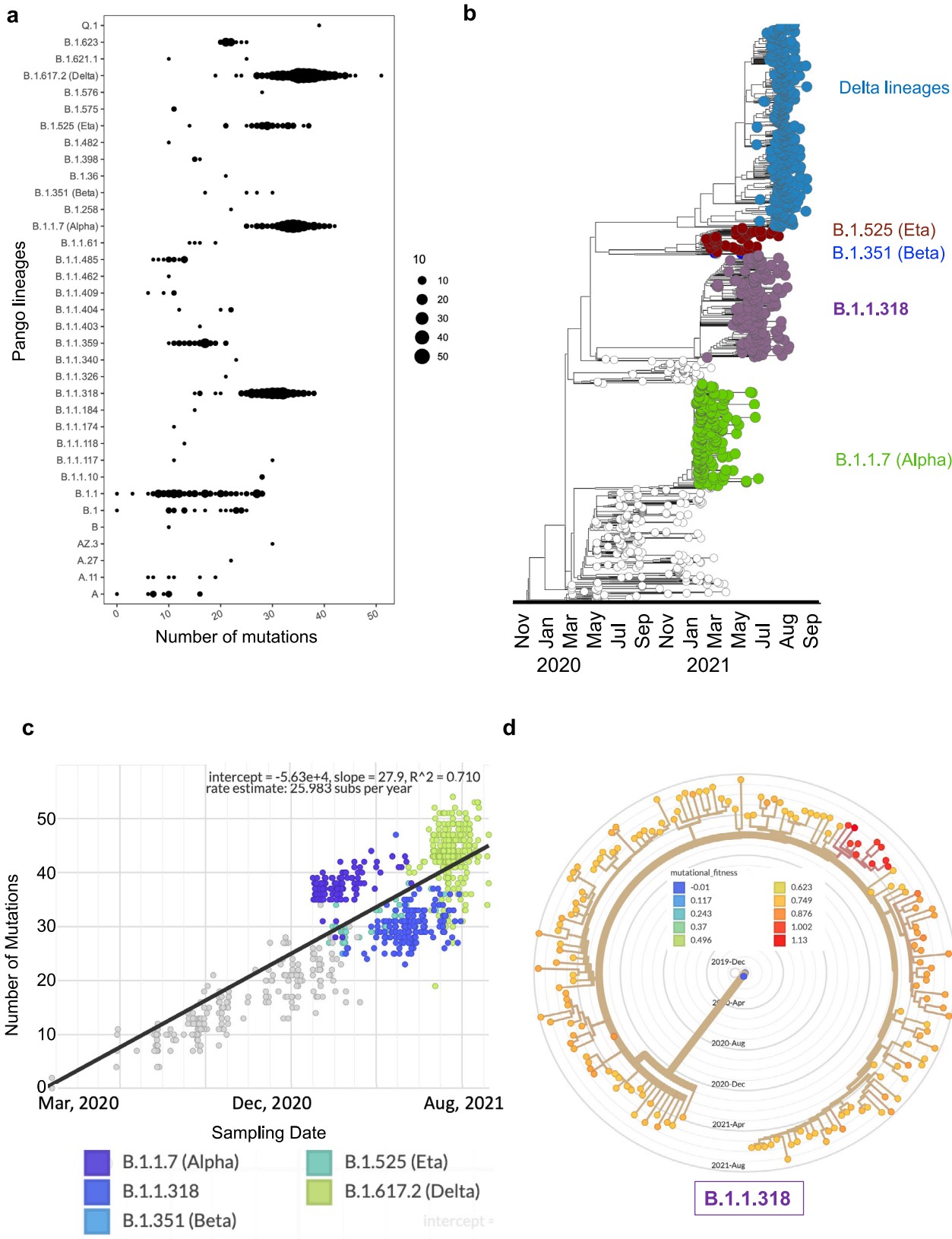

significantly increase the number of cases[17]. As such, it was not surprising that Delta accounted for over 30% of all the cases of COVID-19 successfully sequenced in the current study. This is consistent with data from India, where Delta was first detected in clinical cases and was responsible for many COVID-19 case fatalities in that country[18,19]. Indeed, Delta lineages have been

shown to increase COVID-19 virulence and poor prognosis in certain populations[20]. Unlike most previous variants, nearly all available vaccines have demonstrated limited efficacy against Delta lineages[21], but this trend may also be population-specific since other studies have shown high efficacy[21,22]. Despite very low vaccinations rates and the dominance of Delta lineages, Case

**Fig. 3 Genetic diversity and molecular evolutionary relationships of variants identified in Ghana. a** The spread/range and magnitude of mutation per lineage ($n = 1002$). Each filled circle represents a sample, and the circle's width is proportional to the number of mutations present in a particular sample. **b** Maximum likelihood phylogenetic tree with ancestral state reconstruction in a backdrop of reference sequences from Wuhan and evolutionary relationship of the Ghanaian variants over time ($n = 1002$). Colours show the VOC; Delta lineages, Alpha, B.1.1.318, and Beta, based on Nextstrain's emerging lineages designations. **c** Root-to-tip divergence as a function of sampling time. The Y-axis denoted divergence (the number of mutations in the genome relative to the root), and the X-axis shows the sampling date of each genome. **d** Annotated mutational fitness of all the B.1.1.318 lineage in Ghana; the samples with the highest fitness are coloured red.

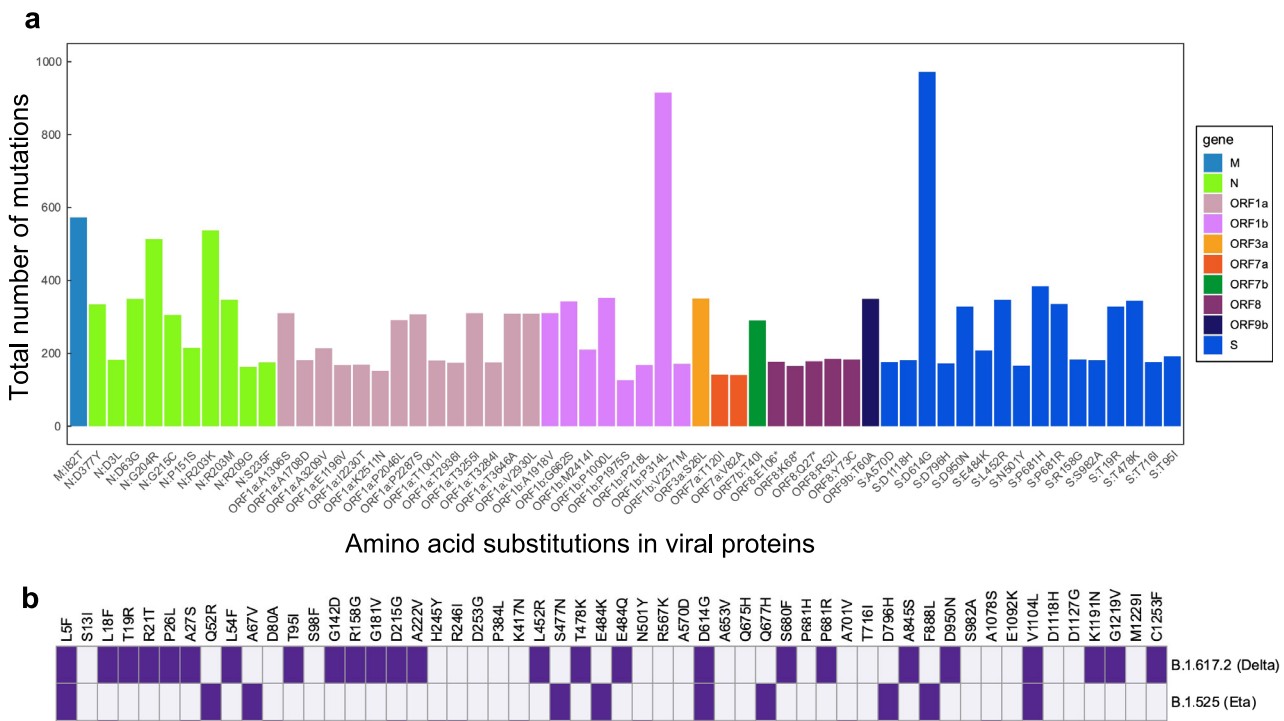

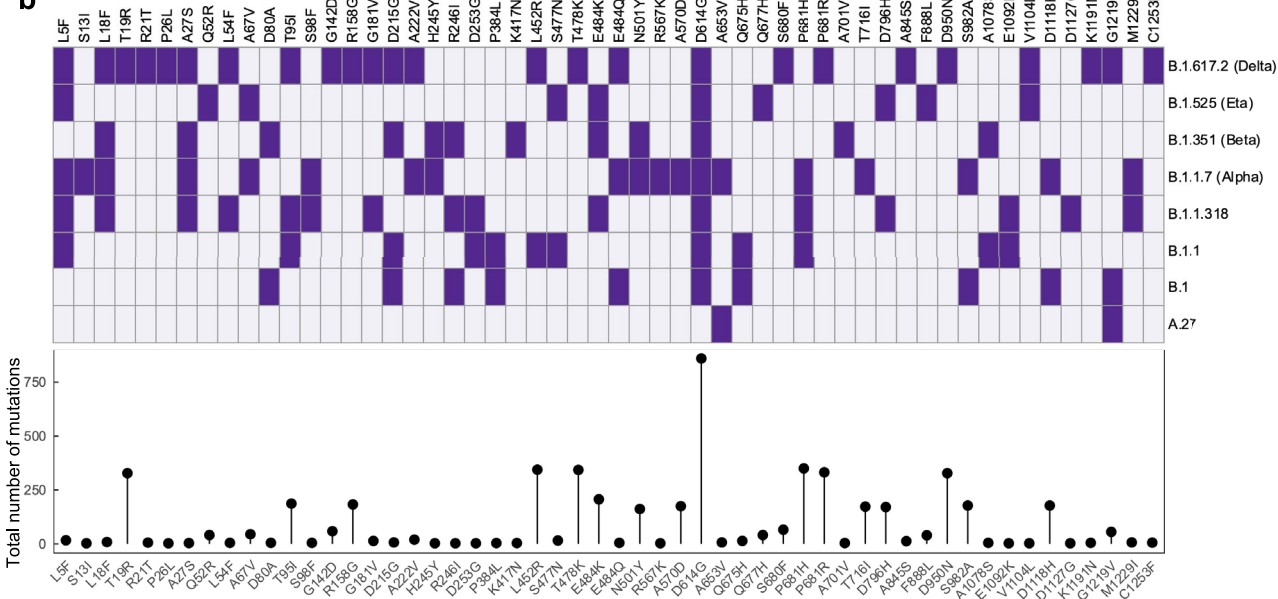

**Fig. 4 Analysis of amino acid substitutions in variants circulating in Ghana. a** Frequencies of amino acid substitutions across all the SARS-CoV-2 proteins in all genomes that were sequenced. The mutations are sorted and coloured per gene. **b** Spike glycoprotein amino acid substitutions present/absent in major SARS-CoV-2 variants circulating in Ghana. Purple shading indicates substitutions within the lineages, whereas white indicates the absence of the substitution in a particular variant in all the samples. The bottom panel plot shows the frequencies of individual substitutions across all the samples.

Fatality Rates remain low to date, a further indication of the apparent resilience of the Ghanaian population to COVID-19 highlighted in our previous study[5]. Nevertheless, as a limitation to the study, the dominance of the VOC in our study may result from the purposive sampling and potential underrepresentation of some variants in other regions.

Through continuous genomic surveillance, we have characterised the diversity and evolution of COVID-19 variants in Ghana. We observed high variation in the number of mutations between samples, suggesting evolution and multiple independent emergences of the Ghanaian variants against the backdrop of a multi-ethnic society. Besides, high mutation frequency within a

population leads to higher chances of diverse variants. Although our study has the limitation of relying on purposive sampling and lack of clinical/epidemiological data such as symptoms and co-morbidities, the large sample size and the depth of genetic analysis performed give high confidence that these data are robust and provide a credible overview of the evolution of the pandemic in Ghana. Enhancing and speeding up vaccinations should be a priority, as well as the pursuit of therapeutic options. Ongoing virus and virus-host interaction experiments, combined with enhanced studies on severe/critically ill patients, should also give more insights into the pathogenesis of SARS-CoV-2 in Ghana.

## Methods

**Study participants**. The study was approved by the Ethics Review Committee of Ghana Health Service (GHS-ERC 005/06/20), the Ethical Committee of the College of Basic and Applied Sciences of the University of Ghana (ECBAS 063/19-20) and the Research Ethics Committee (REC) of the University of Health and Allied Sciences (UHAS) with certificate number UHAS-REC WV [1] 21-22. Samples were obtained from individuals reporting to community COVID-19 testing laboratories in different regions (Ashanti, Bono East, Central, Eastern, Greater Accra, Northern, Upper East, Volta (both Volta and Oti) and Western) of the country, and travellers who tested positive for COVID-19 on arrival at the KIA. Individuals tested at the community laboratories included; patients reporting to hospitals, or those taking tests for travel purposes. The samples from passengers that arrived in Ghana by air were collected at four-time points (March 2020, January 2021, March 2021, and June 2021), while the community samples were continuously collected from the testing laboratories from March 2020 to August 2021. Informed consent was obtained from patients directly or through Ghana Health Service authorisation for samples obtained as part of routine surveillance. All samples were sequenced at West African Centre for Cell Biology of Infectious Pathogens, University of Ghana, except some of the samples from the Volta Region, which were collected and sequenced at the UHAS COVID-19 Testing and Research Centre.

**Sample selection and processing**. Samples confirmed as SARS-COV-2 positive by Real-Time PCR were selected for genome sequencing. Viral RNA from naso-pharyngeal and oropharyngeal samples was extracted using the QIAmp Viral RNA extraction kit (Qiagen, Hilden, Germany). The extracted total RNA concentration was measured using Qubit™ RNA HS Assay Kit on a Qubit 4 Fluorometer (Thermo Fisher Scientific™, MA USA). The integrity and quality of RNA were checked using the Agilent RNA 6000 Nano Kit on the Bioanalyzer (Agilent™ Tech. Inc. CA USA). The ARTIC LoCost protocol (https://artic.network/ncov-2019) was used for sequencing (Extended Methods) as follows; the extracted RNA was converted into cDNA using the LunaScript® RT SuperMix kit (New England Biolabs, UK). The ARTIC V3 primer pools and Q5® Hot Start High-Fidelity DNA polymerase (New England Biolabs, UK) were used for multiplex tiled PCR to generate overlapping amplicons from the cDNA as per the protocol. Sequencing libraries were prepared by end preparation of the amplicons using the NEBNext Ultra II End Repair/dA-tailing module (New England Biolabs, UK) and afterwards barcoded using the EXP-NBD196 kit (Oxford Nanopore Technologies, UK) or the Blunt/TA Ligase Master Mix (New England Biolabs, UK). The barcoded amplicons were then pooled and purified using Ampure XP beads (Beckman Coulter). The purified barcoded library was quantified using the Qubit™ DNA HS Assay Kit (Thermo Fisher Scientific™, USA) with about 75 ng of barcoded libraries ligated to the AMII sequencing adaptors (Oxford Nanopore Technologies, UK) using the Quick ligation kit (New England Biolabs, UK). The adaptor-ligated library was finally purified and quantified using Ampure XP beads (Beckman Coulter) and Qubit™ DNA HS Assay Kit (Thermo Fisher Scientific™, USA) respectively. About 20 ng of the purified adaptor-ligated libraries were loaded on an R9.4.1 flow cell (FLO-MIN106). The sequencing was carried out using a MinION Mk1b or the Mk1c device (Oxford Nanopore Technologies, UK). Our previously published data from March to May 2020 were included in the study to help provide continuity to SARS-Cov-2 genomic epidemiology in Ghana[9].

**Generation of SARS-CoV-2 genomes**. Base-calling and demultiplexing of MinION Fast5 files were performed using Guppy (from version 3.4.3–5.0.7) according to the ARTIC bioinformatic protocols[23]. Sequencing QC was assessed using pycoQC, and demultiplexed reads were aggregated and length filtered using ARTIC guppyplex for a minimum of 400 reads and a maximum of 700 reads to remove chimeric reads. Read QC was assessed using NanoPlot before read alignment, variant calling and consensus generations using ARTIC MinION software (ARTIC version 1.2.1). Alignment metrics, amplicon coverage analysis, variant annotation, and consensus assessment were performed using samtools, mosdepth, BCFtools, SnpEFF, and Quast according to the nfcore/viralrecon pipeline (version 2.2)[24,25]. Variant annotation, validation and quality assessment of the consensus sequence were performed using Viral Annotation DefineR (version 1.1.3)[26]. Genomes that pass quality control were deposited on GISAID and ENA. Phylo-genetic assignment of the consensus sequence to the globally named outbreak

lineages was performed using Pangolin (Versions: pangolin-3.1.14, pangoLEARN 2021-10-13, Pango-designation-1.2.86) according to Rambaut, Holmes[27]. A cov-erage map was generated by comparing the Ghanaian SARS-CoV-2 genomes and the reference genome (Wuhan-Hu-1/2019) using Nextclade CLI (version 1.4.0). This web-based tool performs banded Smith-Waterman alignment with an affine gap-penalty[28]. Nextclade was also used to perform clade assignment and overall quality assessment of the genomes. Further analysis was executed in R (version 4.0.4). Contingency tables were constructed using summary statistics and a $p$ value <0.05 was considered statistically significant.

**Phylogenetic analysis**. Phylogenetic analysis of the SARS-CoV-2 genomes was performed using the Nextstrain pipelines (v11)[28]. Briefly, the nextstrain pipeline incorporates various quality control processes, including; validation of the clinical metadata, aligning sequences using nextalign to identify gaps compared to the SARS-CoV-2 reference genome (MN908947.3 and LR757998.1), and performing pangolin to assign lineages labels[28]. Next, 100 bp from the start and 50 bp at the end were masked, and the regions prone to sequencing errors (13402, 24389 and 24390)[28]. An initial maximum likelihood phylogenetic tree was constructed using augur's fast and stochastic algorithm (IQTREE)[29] with a generalised time-reversible substitution model. This tree was refined to estimate divergence, time, and node dates using a coalescent timescale, then exported to auspice and R (version 4.0.4) for visualisation. Molecular clock estimation for all the lineages and mutational fitness analysis for the B.1.1.318 lineage was also performed using nextstrain pipelines (v11). The mutational fitness was based on hierarchical Bayesian multinomial logistic regression[30].

**Reporting summary**. Further information on research design is available in the Nature Research Reporting Summary linked to this article.

## Data availability

The raw sequencing data ($n = 1077$) is available on the European Nucleotide Archive (PRJEB49489). Our previous data[9] ($n = 46$) are also available on GenBank with accession numbers ranging from MT890204–MT890249. The sequences and basic metadata information such as age, gender, location and sample collection date is freely available at GISAID (https://www.gisaid.org). All accession numbers for GISAID, GenBank, and European Nucleotide Archive are provided in Supplementary Data 1. The summary/analysis tables used to generate the piecharts/stacked plots are available at https://github.com/misita-falcon/SARS-CoV-2-Manuscript-2021 (https://doi.org/10.5281/zenodo.6375851). The WHO dataset is available on the WHO website (https://covid19.who.int/WHO-COVID-19-global-data.csv). The reference genomes (GenBank number: MN908947.3 and LR757998.1) were obtained from Nextrain Github page (https://github.com/nextstrain/ncov/blob/master/data/references_sequences.fasta), but can also be obtained from GenBank (https://www.ncbi.nlm.nih.gov/nuccore/MN908947.3, https://www.ncbi.nlm.nih.gov/nuccore/LR757998.1). Population and Housing Census data were obtained from the Ghana Statistical Service (GSS) (https://www.statsghana.gov.gh) and the map was downloaded from the Ghana Open Data Initiative (https://data.gov.gh/dataset/shapefiles-all-districts-ghana-2012-216-districts).

## Code availability

The scripts used for the analysis reported in this study are publicly available at https://github.com/misita-falcon/SARS-CoV-2-Manuscript-2021 (https://doi.org/10.5281/zenodo.6375851).

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

## Acknowledgements

The study was facilitated by the Government of Ghana through the Ghana Health Service. ARTIC donated the primers used for the study. All data storage and analyses were performed on Zuputo®, the University of Ghana's high-performance computing cluster. We acknowledge the Community and Public engagement members; Kyerewaa A. Boateng and Simon Donkor, Andrew M. Nantogmah and the entire WACCBIP and UHAS COVID-19 Teams. The study was funded by a grant from the Rockefeller Foundation (2021 HTH 006), an Institut de Recherche pour le Développement (IRD) grant (ARIACOV), African Research Universities Alliance (ARUA) Vaccine Development Hubs grant with funds from Open Society Foundation, a Wellcome/African Academy of Sciences Developing Excellence in Leadership Training and Science (DEL-TAS) grant (DEL-15-007 and 107755/Z/15/Z: G.A.A.); National Institute of Health Research (NIHR) (17.63.91) grants using UK aid from the UK Government for a global health research group for Genomic surveillance of malaria in West Africa (Wellcome Sanger Institute, UK) and global research unit for Tackling Infections to Benefit Africa (TIBA partnership, University of Edinburgh); and the World Bank African Centres of Excellence Impact grant (WACCBIP-NCDs: G.A.A.). C.M.M. and D.S.Y.A. are supported by WACCBIP DELTAS PhD fellowships, while P.Q., and Y.B. are supported by a Crick African Network Career Accelerator fellowship. The University of Health and Allied Sciences, COVID-19 Testing and Research Centre was supported by China Novartis Institutes of Biomedical Research (CNIBR-SCD: K.O.D.) and the Grand Challenges Africa programme grant GCA/AMR/rnd2/138: K.O.D.). The views expressed in this publication are those of the author(s) and not necessarily those of the funders.

## Author contributions

G.A.A., P.K.Q., Y.B., and K.O.D. conceived the study. G.A.A., K.O.D., and N.T.N. obtained the funding, and G.A.A., P.K.A., J.O.G. and W.K.A. supervised the work. P.K.Q., K.O.D. and L.N.A. supervised the sequencing and data analyses. J.M.N. performed the viral RNA extraction and sequencing along with F.T.-M., E.B.Q., B.K.Y., D.N.A.M., I.O.W., S.S., V.M., K.T., J.Q., P.C.O., J.G. and R.A.C. B.D.N., I.A.A., P.O.O., S.A., D.A.G., N.A., J.O.C., M.O., A.S., E.D.F., R.A.D., P.T.A., A.K.A., S.K.A., E.A., F.K.A., O.D.B., D.K.M., T.O., L.O.B., E.A.C., S.D., V.As., K.P.A., R.O.P., M.Y.O.A., and P.K.A. contributed to the detection and selection of SARS-CoV-2 positive samples for sequencing. Sequence analysis and generation of figures were performed by C.M.M., D.S.Y.A., V.Ap., E.K.A., P.M.S., F.D., V.V.M. and I.T.O. D.S.Y.A., C.M.M., J.M.N., P.M.S. and V.V.M. drafted the manuscript. L.N.A., P.K.Q. and G.A.A. critically reviewed and edited the manuscript. All authors read and approved the final version of the manuscript.

## Competing interests

The authors declare no competing interests.

## Additional information

[1]West African Centre for Cell Biology of Infectious Pathogens (WACCBIP), University of Ghana, Accra, Ghana. [2]Department of Biochemistry, Cell and Molecular Biology, University of Ghana, Accra, Ghana. [3]University of Health and Allied Sciences COVID-19 Testing and Research Centre, Ho, Ghana. [4]Tamale Teaching Hospital Intensive Care Unit, Ghana Health Service, Tamale, Ghana. [5]Noguchi Memorial Institute for Medical Research, University of Ghana, Legon, Accra, Ghana. [6]LEDing Medical Laboratory, Accra, Ghana. [7]Biomedical and Public Health Research Unit,

Council for Scientific and Industrial Research, Accra, Ghana. [8]Kintampo Health Research Centre, Research and Development Division, Ghana Health Service, Kintampo North Municipality, Ghana. [9]Ghana Infectious Disease Centre, Accra, Ghana. [10]Kumasi Centre for Collaborative Research in Tropical Medicine, Kwame Nkrumah University of Science and Technology, Kumasi, Ghana. [11]Accra Veterinary Laboratory, Veterinary Services Directorate, Accra, Ghana. [12]Tamale Public Health and Reference Laboratory, Ghana Health Service, Tamale, Ghana. [13]Institutional Care Division (ICD), Ghana Health Service, Accra, Ghana. [14]Cape Coast Teaching Hospital, Ghana Health Service, Cape Coast, Ghana. [15]Takoradi Veterinary Services Department, Ghana Health Service, Takoradi, Ghana. [16]Ga East Municipal Hospital, Ghana Health Service, Accra, Ghana. [17]Pong-Tamale Central Veterinary Laboratory, National Veterinary Services Directorate, Tamale, Ghana. [18]Navrongo Health Research Centre, Research and Development Division, Ghana Health Service, Navrongo, Ghana. [19]Ghana Health Service, Accra, Ghana. [20]Institut de Recherche pour le Développement, Marseille, France. [21]Yemaachi Biotechnology, Accra, Ghana. ✉email: pquashie@ug.edu.gh; lamenga-etego@ug.edu.gh; gawandare@ug.edu.gh

