## [Peer Review File · Nature Communications]

Genetic diversity of SARS-CoV-2 infections in Ghana from 2020-2021REVIEWER COMMENTS

Reviewer #1 (Remarks to the Author):

The authors sequenced around a thousand SARS-CoV-2 genomes from Ghana and present lineages, phylogenetic relationships, and mutational patterns. The paper is a routine investigation of SARS-CoV-2 lineages from 2020 till summer of 2021. Figures should be improved in quality and clarity. Authors should at least try to see whether there is a link between mutations or lineages circulating and epidemiological data from the cases. Such analysis would have made the paper stronger.

“With the highest number of active COVID-19 cases, the Greater Accra region had the highest number of sequenced samples.” Is this region more populated than others? Can authors replace the map in figure 1 with a map with population density per region and sampling? Also, where the 104 arrived by airplane (showed in the figure) arrived in the map? Authors did not explain in the figure legend what are the grayscales representing in the map. Panels c and b are ok, but it would be more helpful to add also a temporal dimension to the data, by month. I would encourage the authors on keeping with the same colors between panels B, C and D. It's very confusing to see lineages across panels represented by different colors.

“11.7 % (n=106) were from travellers arriving in the country through the Kotoka International Airport” where the travellers arrived from?

This sentence is unclear: “Interestingly, the B.1. 1, B.1.359, B.1.1., and B.1.623 that dominated Ghana in 2020 became supplanted by the modal variants responsible for most transmissions in all the regions.” Can authors explain what modal variants are? What is the basis for this sentence, how do the authors know that these variants were responsible for most transmissions?

“VOCs were detected in travellers from several of Ghana's neighbouring countries,” add a table with countries and number of introductions and timing of introduction and refer to the table here in this sentence.

“Interestingly, the Beta and Kappa variants did not become dominant in Ghana; instead, B.1.1.318, which is likely to have originated from Nigeria, and detected in a traveller from Gabon, became dominant in Ghana.” Do authors have any hypothesis on why this is?

How panel a in figure 2 differ from panel b in figure 1? I suppose figure 1 shows the total of the genomes sampled. Is this containing the 106 shown in panel a? Figure legends need to be more detailed. Again, I would encourage the authors to be consistent with colors for lineages across figures to allow comparisons. How is the map in figure 2 done? Are the links based on the travel reported, or is it a Bayesian phylogeographic analysis?

Do authors have the Ct values for the samples? Do they correlate with symptoms?

“Using a phylogenetic tree, we outline the transmission events,” I think this statement is incorrect. Merely based on phylogenetic trees is not possible to outline transmission events, but only phylogenetic relationships. Without contact tracing data it's not possible to infer directionality of infection and/or transmission.

Can authors show the frequency of the mutations over time? Is there any particular trend?

“The Eta variant had the highest (three) individual mutations (Q52R, Q677H and F888L) in the spike protein compared to other VOCs, contributing to its adaptability in Africa.” How do the authors know that these mutations are the ones contributing to its adaptability in Africa?

Reviewer #2 (Remarks to the Author):

In this manuscript, the authors describe a study, where they sequenced over 1000 cases of SARS-CoV-2 in Ghana. The study is interesting and enriches the literature of the SARS-CoV-2 evolution in West Africa.

Major comments

However, it presents a sampling bias limitation. The Greater Accra region constitutes almost 50% of sequenced samples (Fig. 1a). It would be valuable to see from the authors a supplementary figure that shows the number of positive cases (out of total tested) per region per month during the study period and align that with the ratio of samples sequenced per region.

Additional to percentage of variants and regions, authors should add the component of time (months covered) for Fig. 1c and Fig. 1d, to clearly depict the variation of lineages over time.

Lines 181-182 refer to cases infected with Delta and Delta-Plus being associated with severe/critical COVID-19 cases. There should be additional information on these cases such as age and co-morbidities (if any). There should be a supplementary figure that reinforces Fig. 3d in a sense that age, and co-morbidities metrics should be captured as well.

Numerous studies have demonstrated repeated introductions of SARS-CoV-2 into different places. This study would markedly benefit from demonstrating the proportion of cases in Ghana caused by introductions from elsewhere and their impact on transmissibility of the virus locally

Minor comments

In their methodology, authors selected for genomic sequencing samples confirmed as SARS-COV-2 positive by Real-Time PCR with cycle threshold Ct- values in the range of 12–35. Most of samples with Ct- value >30 do not pass quality check. Authors should revise this and provide a supplementary table showing samples which were eligible for genomic sequencing for more clarity.

Reviewer #3 (Remarks to the Author):

The data presented in this paper is important in understanding evolution of SARS-CoV-2 epidemic in Ghana and more generally in West Africa during the first one and half years. Using positive samples collected from multiple regions and for different purposes including traveler surveillance, the analysis very clearly demonstrates patterns of lineage introduction and replacement in Ghana. However, the sub-analysis and conclusions on clinical presentation, and of local evolution and sources of introduction require more rigor as no formal & robust comparison of Ghanaian SARS-CoV-2 genomes and the rest of the world is presented. Additional comments below:

Line 20: B.1.1 lineage

Line 23: (a) Delta (B.1.617.2), (b) avoid use of Delta plus, it has an ambiguous meaning, sometimes refers to AY.4.2 or AY.1

<https://www.gavi.org/vaccineswork/theres-now-delta-plus-variant-covid-19-what-does-mean>

https://assets.publishing.service.gov.uk/government/uploads/system/uploads/attachment_data/file/993879/Variants_of_Concern_VOC_Technical_Briefing_15.pdf

Line 25: Is this not the case globally as VoCs spread? Were any of the mutations unique to Ghana?

Line 55: Current knowledge is that mutations arise only during replication due to error prone RdRp, and these are then acted upon by selection forces within host the or/and during person-to-person transmission

Line 58 and throughout the manuscript (use of terminology): need to distinguish mutations (these affect nucleotide sequences) from amino acid substitutions/ changes. AA changes arise from occurrence of non-synonymous mutations.

Line 61: what is social measures?

Line 91: No need to separate B.1.617.2 and AY.* lineages, just combine and say "Delta lineages"

Line 98 and throughout the manuscript: No need to provide percentages to 1dp, the denominators are generally small numbers, thus such precision is unwarranted.

Line 101: With the small numbers, sometime it is difficult to access if a lineage is dominating especially when cases are also identified from contact tracing which creates bias especially when the denominator is a small number.

Line 112: There are typos in this line affecting the lineage names

Line 113: what do you mean by "modal variant"?

Line 115: Not all VoCs have been detected in Ghana? Gamma is not in this data

Figure 1b and 2a: Pango lineages, not "Pangolin"

Line 132: List the neighboring countries

Line 138: what is the evidence that B.1.1318 was probably imported from Nigeria? Did the lineage arrive Ghana just once? Single or multiple importation? Why not from Mauritius via another country?

Figure 2c: Are there instances where multiple lineages were identified from travellers in the same country? (The resolution of the map is quite low, can see the W/Africa countries clearly, consider a zoomed in version as well)

Line 150: (For all figures applicable), I encourage the authors to include in the data from March-June 2020/ Ngoi et al

Line 171: Symptom status can change during a SARS-CoV-2 infection. Is the analysis here based on a spot check at the time of sampling alone? If so, is this analysis valid? Further, this result will be influenced by the sampling criteria for the genome sequenced from the total positives. These caveats need to be recognized.

Present a table summarizing the demographic characteristics of the sequenced versus non-sequenced cases (age, sex, region, symptom status and reason for testing and vaccination status)

Line 204: The statement assumes that lineages had a single introduction followed by local diversification which is very unlikely. Was multiple introductions of diverse viruses within the same lineage ruled out?

Line 207: rephrase use of word "exciting"; Line 209: It is difficult to confirm transmission events from genomic data alone; line 213, where is this result? Line 221: not sure how you define mutational fitness.

Figure 4b. Include the earlier genomes from Ngoi et al; Figure 5a, sort x-axis by gene/protein
There is little-to-no like of the current result to previous data from the team (Ngoi et al) which can help provide continuity to SARS-Cov-2 genomic epidemiology in Ghana

Line 268: Can not see the lag for wave 1 relative to the rest of Africa

Line 315: Due to potential biases in the way the data here is collected, the finding of analysis on disease severity patterns here are challenging to conclude. There are multiple factors that require adjusting for age, co-morbidities, reason for testing etc

Line 330: Not sure that evidence for rapid evolution comes from this analysis

Line 361: How many samples were availed for analysis, how many were processed for WGS and how many never made it to the lineage assignment analysis? Provide details of Kits and conditions used cDNA synthesis, PCR and library preparation. Details of sequencing strategy/platform are lacking. The ref 23 does not seem right.

Line 382: include Pango and pangoLearn version

Line 390: Nextstrain

Title: The surveillance has lasted > 1 year

Manuscript title: Genetic diversity of SARS-CoV-2 infections in Ghana from 2020-2021

Responses to reviewer comments

Reviewer #1

Comment: The authors sequenced around a thousand SARS-CoV-2 genomes from Ghana and present lineages, phylogenetic relationships, and mutational patterns. The paper is a routine investigation of SARS-CoV-2 lineages from 2020 till summer of 2021.

Comment: Figures should be improved in quality and clarity.

Response: All Figures have been revised to improve clarity and color coding in high quality jpeg format.

Comment: Authors should at least try to see whether there is a link between mutations or lineages circulating and epidemiological data from the cases. Such analysis would have made the paper stronger.

Response: It must be noted that because the samples were collected in pandemic context, the epidemiological data that was collected along with the samples is limited. But we have added the demographic description of the patients, such as age category, sex, and sampling location (Supplementary table 3). Furthermore, a multivariate logistic regression analysis did not reveal any significant relationship between the demographic characteristics of the participant and the type of lineage detected.

Comment: “With the highest number of active COVID-19 cases, the Greater Accra region had the highest number of sequenced samples” Is this region more populated than others?

Response: The 2021 Ghana Population and Housing Census data shows Greater Accra as the region with highest population of about 5.4 million people and the highest population density by far. In addition, the Greater Accra region has Ghana’s only international airport, with majority of arriving travelers staying in Accra and likely contributing to the introduction of new infections. The population profile of all the regions in Ghana and the COVID-19 cases per region are highlighted in the new Figure 1a and supplementary Table 1 in the revised manuscript. (<http://www.statsghana.gov.gh>) (<https://www.ghs.gov.gh/covid19/dashboardm.php>)

Comment: Can authors replace the map in figure 1 with a map with population density per region and sampling? Also, where the 104 arrived by airplane (showed in the figure) arrived in the map? Authors did not explain in the figure legend what are is grayscale representing in the map.

Response: The Fig 1a map has been replaced with a map showing the population density per region and sampling. The individuals arrived at the Kotoka International Airport, which is in the Greater Accra Region. The countries of origin and the time of arrival are shown in Table 1. The number of COVID-19 cases are also shown per region as well as the number of samples sequenced in Supplementary Table 1. The grayscale has been replaced with the population density per hectare (colored in blue) and explained in the figure legend.

Comment: Panels c and b are ok, but it would be more helpful to add also a temporal dimension to the data, by month. I would encourage the authors on keeping with the same colors between panels B, C and D. it’s very confusing to see lineages across panels represented by different colors.

Response: We thank the reviewer for this important observation on panel B, C and D. The temporal dimension of the data was presented in Fig 2b and 2c for the community samples and 2021 community samples respectively. The color coding for all the lineages have been resolved and the same colors maintained across all the figures.

Comment: “11.7 % (n=106) were from travellers arriving in the country through the Kotoka International Airport” where the travellers arrived from?

Response: We have included a Table 1 that shows the self-reported origin of 121 travelers, the month they arrived at Kotoka international Airport, and the variants that was detected. The number of travelers has increased from 106 after adding the 46 genomes from Ngoi et al (which contained 15 more traveler samples)

Comment: This sentence is unclear: “Interestingly, the B.1. 1, B.1.359, B.1.1., and B.1.623 that dominated Ghana in 2020 became supplanted by the modal variants responsible for most transmissions in all the regions.” Can authors explain what modal variants are? What is the basis for this sentence, how the authors know that these variants where responsible for most transmissions?

Response: We agree with the reviewer that this sentence was unclear, and the sentence has been changed to " The B.1, B.1.1.359, B.1.1, and B.1.623 that dominated Ghana in 2020 became supplanted by Alpha and Delta VOCs in most of the regions”

Comment: “VOCs were detected in travellers from several of Ghana's neighbouring countries,” add a table with countries and number of introductions and timing of introduction and refer to the table here in this sentence.

Response: A table showing countries from which all the variants were introduced into Ghana as well as the month and year of introduction has been included as Table 1.

Comment: “Interestingly, the Beta and Kappa variants did not become dominant in Ghana; instead, B.1.1.318, which is likely to have originated from Nigeria, and detected in a traveller from Gabon, became dominant in Ghana.” Do authors have any hypothesis on why this is?

Response: The sentence has been changed to “Interestingly, the Beta and Kappa variants did not become dominant in Ghana; instead, B.1.1.318, which was detected in travellers from Nigeria, Gabon, and Dubai, became dominant in Ghana.” Our hypothesis is that these variants had lower fitness or were less transmissible than B.1.1.318 and Alpha.

Comment: How panel a in figure 2 differ from panel b in figure 1? I suppose figure 1 shows the total of the genomes sampled. Is this containing the 106 shown in panel a? figure legends need to be more detailed. Again, I would encourage the authors to be consistent with colors for lineages across figures to allow comparisons.

Response: We have included more details in all figure legends. Figure 1b shows total genomes sampled including the travelers (n=1123). The original figure 2 has been replaced with Table 1, which shows data for only travelers (n=121). The colors have also been made consistent accordingly.

Comment: How is the map in figure 2 done? Are the links based on the travel reported, or is it a Bayesian phylogeographic analysis?

Response: The map links was generated based on the reported travel history. The map has been replaced by Table 1 for more clarity.

Comment: Do authors have the ct values for the samples? Do they correlate with symptoms?

Response: The analysis on clinical symptoms has been removed because of possible sampling bias as suggested by reviewer 3.

Comment: “Using a phylogenetic tree, we outline the transmission events,” I think this statement is incorrect. Merely based on phylogenetic trees is not possible to outline transmission events, but only phylogenetic relationships. Without contact tracing data it’s not possible to infer directionality of infection and/or transmission.

Response: We are grateful to the reviewer for the comments, the statement has been corrected accordingly to reflect phylogenetic relationships.

Comment: Can authors show the frequency of the mutations over time? Is there any particular trend?

Response: The frequency of mutations over time has been shown in Figure 3c and the trend is that the number of mutations increased over time. The VOCs in Ghana tended to have more mutations compared to the other lineages.

Comment: “The Eta variant had the highest (three) individual mutations (Q52R, Q677H and F888L) in the spike protein compared to other VOCs, contributing to its adaptability in Africa.” How the authors know that these mutations are the ones contributing to its adaptability in Africa?

Response: This is a valid question, and we have accordingly revised the statement to read the “Eta variant had the highest (three) individual mutations (Q52R, Q677H and F888L) in the spike protein compared to other VOCs, possibly contributing to its adaptability in the Ghanaian population”.

Reviewer #2 (Remarks to the Author):

Comment: In this manuscript, the authors describe a study, where they sequenced over 1000 cases of SARS-CoV-2 in Ghana. The study is interesting and enriches the literature of the SARS-CoV-2 evolution in West Africa.

Major comments

Comment: However, it presents a sampling bias limitation. The Greater Accra region constitutes almost 50% of sequenced samples (Fig. 1a). It would be valuable to see from the authors a supplementary figure that shows the number of positive cases (out of total tested) per region per month during the study period and align that with the ratio of samples sequenced per region.

Response: We have provided a Supplementary Table (1), outlining the population in each region, the number of positive cases per region as well as the number of samples sequenced. Since we do not perform COVID-19 testing services in our facilities, we rely on the testing labs to share samples and we sequence all that we can get. The table shows that Greater Accra has the highest population in Ghana (> 5 million) people in a comparatively much smaller land area, giving a comparatively higher population density. The region contributed the highest number of positive cases (61 %) compared to the other regions. Therefore, the high proportion of samples sequenced from Great Accra is a reflection of the fact that vast majority of cases in Ghana occur in _____ that _____ region.

Comment: Additional to percentage of variants and regions, authors should add the component of time (months covered) for Fig. 1c and Fig. 1d, to clearly depict the variation of lineages over time.

Response: The component of time is shown in Fig 2b and 2c and clearly depicts the variation of lineages over time from Mar 2020 to Sep 2021.

Comment: Lines 181-182 refer to cases infected with Delta and Delta-Plus being associated with severe/critical COVID-19 cases. There should be additional information on these cases such as age and co-morbidities (if any). There should be a supplementary figure that reinforces Fig. 3d in a sense that age, and co-morbidities metrics should be captured as well.

Response: The data on the disease severity has been removed as suggested by reviewer 3.

Comment: Numerous studies have demonstrated repeated introductions of SARS-CoV-2 into different places. This study would markedly benefit from demonstrating the proportion of cases in Ghana caused by introductions from elsewhere and their impact on transmissibility of the virus locally

Response: Table 1 and supplementary table (4) have been added demonstrating the proportion of travelers, Month of introduction, and their reported country of origin. The table also shows the months in which the cases were introduced and the number of samples with specific variants in that month. In addition, Fig 2c shows the dynamics of VOCs which coincides with the months of introductions.

Minor comments

Comment: In their methodology, authors selected for genomic sequencing samples confirmed as SARS-COV-2 positive by Real-Time PCR with cycle threshold Ct- values in the range of 12–35. Most of samples with Ct- value >30 do not pass quality check. Authors should revise this and

provide a supplementary table showing samples which were eligible for genomic sequencing for more clarity.

Response: The section has been revised, to show that samples with good quality were sequenced. A statement has been added on the results section to highlight the samples that were eligible for genomic sequencing and those that were eligible for lineage assignment.

Reviewer #3 (Remarks to the Author):

Comment: The data presented in this paper is important in understanding evolution of SARS-CoV-2 epidemic in Ghana and more generally in West Africa during the first one and half years. Using positive samples collected from multiple regions and for different purposes including traveler surveillance, the analysis very clearly demonstrates patterns of lineage introduction and replacement in Ghana.

Comment: However, the sub-analysis and conclusions on clinical presentation, and of local evolution and sources of introduction require more rigor as no formal & robust comparison of Ghanaian SAR-CoV-2 genomes and the rest of the world is presented.

Response: We agree with the reviewer that the analysis can be improved, and we have accordingly provided additional sub-analysis with summary data in Table 1 and Supplementary Table 4.

Additional comments below:

Comment: Line 20: B.1.1 lineage

Response: The correction has been made accordingly

Comment: Line 23: (a) Delta (B.1.617.2), (b) avoid use of Delta plus, it has an ambiguous meaning, sometimes refers to AY.4.2 or AY.1 <https://www.gavi.org/vaccineswork/theres-now-delta-plus-variant-covid-19-what-does-mean>

https://assets.publishing.service.gov.uk/government/uploads/system/uploads/attachment_data/file/993879/Variants_of_Concern_VOC_Technical_Briefing_15.pdf

Response: We have changed references to Delta and Delta plus variants to Delta lineages or Delta variants throughout the manuscript

Comment: Line 25: Is this not the case globally as VoCs spread? Were any of the mutations unique to Ghana?

Response: We have modified the statement to “The apparent rapid viral evolution observed demonstrates the potential for emergence of novel variants with greater mutational fitness as observed elsewhere in the world.”

Comment: Line 55: Current knowledge is that mutations arise only during replication due to error prone RdRp, and these are then acted upon by selection forces within host the or/and during person-to-person transmission

Response: We thank the reviewer for pointing this out, we have revised the statement accordingly: “Like other RNA viruses, most mutations in the SARS-CoV-2 genome arise during viral replication, and the resulting mutant viruses are then subjected to selective pressures within the host and/or during inter-person transmission.”

Comment: Line 58 and throughout the manuscript (use of terminology): need to distinguish mutations (these affect nucleotide sequences) from amino acid substitutions/ changes. AA changes arise from occurrence of non-synonymous mutations.

Response: The manuscript has been edited extensively to distinguish the mutations and amino acid changes

Comment: Line 61: what is social measures?

Response: The term “Social measures” has been removed.

Comment: Line 91: No need to separate B.1.617.2 and AY.* lineages, just combine and say “Delta lineages”

Response: The B.1.617.2 and AY.* have been merged and labeled Delta lineages throughout the manuscript.

Comment: Line 98 and throughout the manuscript: No need to provide percentages to 1dp, the denominators are generally small numbers, thus such precision is unwarranted.

Response: All the percentages have been edited accordingly as suggested.

Comment: Line 101: With the small numbers, sometime it is difficult to access if a lineage is dominating especially when cases are also identified from contact tracing which creates bias especially when the denominator is a small number.

Response: This concern is valid. However, the samples come from many different testing labs spread across the regions, so it is unlikely that the impact of contact tracing will be significant in this case. Moreover, the contact tracing in Ghana is not very efficient and not extensive enough to result in samples from the same index case being sent to multiple labs.

Comment: Line 112: There are typos in this line affecting the lineage names

Response: The line has been edited accordingly

Comment: Line 113: what do you mean by “modal variant”?

Response: The term was used to refer to Variants of Concern, but this confusing notation has now been removed through the manuscript and stated clearly as VOCs.

Comment: Line 115: Not all VoCs have been detected in Ghana? Gamma is not in this data

Response: The statement has been edited accordingly to reflect only the VOCs that were detected in Ghana.

Comment: Figure 1b and 2a: Pango lineages, not “Pangolin”

Response: The label has been edited as suggested.

Comment: Line 132: List the neighboring countries

Response: We have now listed the neighbouring countries such as Nigeria, Ivory Coast, and Burkina Faso as suggested

Comment: Line 138: what is the evidence that B.1.1318 was probably imported from Nigeria? Did the lineage arrive Ghana just once? Single or multiple importation? Why not from Mauritius via another country?

Response: We have addressed this question, which was also raised by the reviewer 1. The lineage arrived twice in March (n=3) and June 2021 (n=1), with multiple importations from three countries; Nigeria, Gabon, and Dubai. Thus, the statement has been changed accordingly to state that “Interestingly, the Beta and Kappa variants did not become dominant in Ghana; instead, B.1.1.318, which was detected in travellers from Nigeria, Gabon, and Dubai, became dominant in Ghana.”

Comment: Figure 2c: Are there instances where multiple lineages were identified from travellers in the same country? (The resolution of the map is quite low, can see the W/Africa countries clearly, consider a zoomed in version as well)

Response: We thank the reviewer for the comments, table 1 has been provided, and shows that multiple lineages were introduced from the same country. The Fig has been replaced by Table 1.

Comment: Line 150: (For all figures applicable), I encourage the authors to include in the data from March-June 2020/ Ngoi et al

Response: We have included the data (n=46) from Ngoi et al for the entire analysis, and now our sample size is 1123 instead of 1077.

Comment: Line 171: Symptom status can change during a SARS-CoV-2 infection. Is the analysis here based on a spot check at the time of sampling alone? If so, is this analysis valid? Further, this result will be influenced by the sampling criteria for the genome sequenced from the total positives. These caveats need to be recognized.

Response: This analysis is based on acute symptoms that patients presented with at the time of sampling prior to the initiation of treatment, so this is a consistent measure and a valid comparison across variants. For the avoidance of doubt, this analysis has been removed from the manuscript.

Comment: Present a table summarizing the demographic characteristics of the sequenced versus non-sequenced cases (age, sex, region, symptom status and reason for testing and vaccination status)

Response: Most of our samples are collected as part of the National Surveillance of COVID-19, so the demographic data are stored centrally and only accessed for samples that were successfully sequenced. A supplementary table 3 has been presented on the sequenced cases, to highlight the demographic information such as age categories, sex, and location. The study was conducted mostly when vaccination coverage was very low (~1%), hence the number of vaccinated individuals in the study participants was negligible.

Comment: Line 204: The statement assumes that lineages had a single introduction followed by local diversification which is very unlikely. Was multiple introductions of diverse viruses within the same lineage ruled out?

Response: Yes, multiple introductions are possible as shown in Table 1 and supplementary table 4, therefore the word local has been removed from the statement.

Comment: Line 207: rephrase use of word “exciting”;

Response: We thank the reviewer for the comment, the word has been removed.

Comment: Line 209: It is difficult to confirm transmission events from genomic data alone.

Response: The word transmission events has been replaced with phylogenetic relationships

Comment: Line 213, where is this result?

Response: The result on delta plus variants have been provided in the Supplementary Table 5

Comment: Line 221: not sure how you define mutational fitness.

Response: Mutational fitness is a new analysis method to determine which variants have gained fitness using their mutational profile and time of sampling as implemented on Nextstrain pipelines (v11). The method is based on hierarchical Bayesian multinomial logistic regression [Obermeyer FH, *et al.* Analysis of 2.1 million SARS-CoV-2 genomes identifies mutations associated with transmissibility. *medRxiv*, (2021)]

Comment: Figure 4b. Include the earlier genomes from Ngoi et al;

Response: The 46 genomes have been included and thus the sample size has changed from 1007 to 1123.

Comment: Figure 5a, sort x-axis by gene/protein

Response: The figure has been sorted by protein.

Comment: There is little-to-no like of the current result to previous data from the team (Ngoi et al) which can help provide continuity to SARS-Cov-2 genomic epidemiology in Ghana

Response: The data from previous work has been added to this manuscript to provide continuity of the SARS-Cov-2 genomic epidemiology in Ghana from March 2020 to September 2021.

Comment: Line 268: Cannot see the lag for wave 1 relative to the rest of Africa

Response: The statement has been edited to clarify that it lagged in the second and third wave

Comment: Line 315: Due to potential biases in the way the data here is collected, the finding of analysis on disease severity patterns here are challenging to conclude. There are multiple factors that require adjusting for age, co-morbidities, reason for testing etc

Response: We do acknowledge the underlining biases in the data, lack of well-characterized epidemiological data due to the pandemic situation within which these samples were collected. However, these samples came from patients who tested for COVID because they felt unwell and reported to hospital. But for the avoidance of doubt, this analysis has been removed from the manuscript.

Comment: Line 330: Not sure that evidence for rapid evolution comes from this analysis

Response: The word “rapid” has been removed.

Comment: Line 361: How many samples were availed for analysis; how many were processed for WGS and how many never made it to the lineage assignment analysis?

Response: A statement has been added in the results section as “A total of 2,213 samples were available for whole genome sequencing (WGS), 1,987 samples were processed for WGS, and 1573 samples made it to lineage assignment (Supplementary Table 2)”. The success rate is impacted by the lack of infrastructural capacity in the testing laboratories for storage and proper sample shipment.

Comment: Provide details of Kits and conditions used cDNA synthesis, PCR, and library preparation. Details of sequencing strategy/platform are lacking. The ref 23 does not seem right.

Response: The details have been provided extensively in the methods section under the sample selection and processing

Comment: Line 382: include Pango and pangolearn version

Response: The versions have been added as (Versions: pangolin-3.1.14, pangolearn 2021-10-13, pango-designation-1.2.86)

Comment: Line 390: Nextstrain

Response: The correction has been made.

Comment: Title: The surveillance has lasted > 1 year

Response: The title has been revised to reflect the actual sampling period “**Genetic diversity of SARS-CoV-2 infections in Ghana from 2020-2021**”

REVIEWERS' COMMENTS

Reviewer #1 (Remarks to the Author):

Authors have addressed all the comments.

Reviewer #2 (Remarks to the Author):

I thank the authors for their detailed response, and for the improvements to the paper which have helped clarify the contribution in my mind.

To be specific, I am happy that the authors have addressed my previous queries regarding sampling bias limitations. Although the Supplementary Table 1 mentioned in their rebuttal letter does not match with their responses, I guess this might be a minor error. The Supplementary Table 2 gives enough information in this regard.

At this stage, I feel that the authors have sufficiently addressed the considerations I raised during my initial review. My view is that this paper is an interesting one that could well be worthy of publication in a journal of the standard of Nature Communications.

Reviewer #3 (Remarks to the Author):

My previous concerns about this manuscript have been adequately addressed by the authors.

Responses to reviewer comments

Reviewer #1 (Remarks to the Author):

Comment: Authors have addressed all the comments.

Reviewer #2 (Remarks to the Author):

Comment: I thank the authors for their detailed response, and for the improvements to the paper which have helped clarify the contribution in my mind. To be specific, I am happy that the authors have addressed my previous queries regarding sampling bias limitations. Although the Supplementary Table 1 mentioned in their rebuttal letter does not match with their responses, I guess this might be a minor error. The Supplementary Table 2 gives enough information in this regard.

Response: We thank the reviewer for the comment, the Supplementary Table 1 was an error in the first response letter, and we had intended to reference Supplementary Table 2.

Comment: At this stage, I feel that the authors have sufficiently addressed the considerations I raised during my initial review. My view is that this paper is an interesting one that could well be worthy of publication in a journal of the standard of Nature Communications.

Reviewer #3 (Remarks to the Author):

Comment: My previous concerns about this manuscript have been adequately addressed by the authors.